# TN-AutoRCA: Benchmark Construction and Agentic Framework for Self-Improving Alarm-Based Root Cause Analysis in Telecommunication Networks

## Abstract

Root Cause Analysis (RCA) in Telecommunication Networks is a critical task, yet it presents a formidable challenge for Artificial Intelligence (AI) due to its complex, graph-based reasoning requirements and the scarcity of realistic benchmarks. To catalyze research in this domain, We herein present **TN-RCA**, an inaugural real-world, publicly accessible benchmark for root cause analysis (RCA) of telecommunication network alarms, comprising 530 fault scenarios constructed from expert-validated Knowledge Graphs(KGs). Our evaluation reveals that even state-of-the-art Large Language Models (LLMs) perform poorly on this task, with the best LLMs achieving an F1-score below 70%, highlighting its significant difficulty.To address this challenge, we then propose **Auto-RCA**, a novel agentic system that produces the core code to analysis root cause through the automatically iterative refinement. The core innovation of Auto-RCA lies beyond simple self-correction; it employs an iterative "evaluate-analyze-repair" loop that systematically identifies common patterns across all failure cases to generate contrastive feedback. This feedback guides the LLM to fix systemic logical flaws rather than isolated errors. Experiments show that this agentic framework dramatically boosts the performance of root cause analysis in telecommunication networks, raising the final F1-score on TN-RCA from 58.99% (achieved by Gemini-2.5-Pro directly) to 91.79%. Meanwhile, Auto-RCA achieves 50% accuracy on the OpenRCA Xu et al. (2025) benchmark and demonstrates its generalization. This work not only contributes a crucial benchmark to the community but also demonstrates that autonomous, self-optimizing agentic architecture is a powerful paradigm for solving complex, domain-specific reasoning problems.

## 1 Introduction

Alarm-Based Root Cause Analysis (RCA) is a cornerstone of reliability engineering in telecommunication networks, essential for ensuring network resilience and minimizing service downtime. However, traditional approaches to alarm root cause analysis rely critically on domain experts to perform detailed analysis of intricate alarm log data for accurate root cause identification. This reliance introduces significant subjectivity tied to individual experience, hindering the establishment of unified data management protocols and systematic modeling frameworks. Recent advances in LLMs have yielded remarkable progress in reasoning capabilities, with state-of-the-art models achieving human-level or superior performance on mathematical and coding benchmarks DeepSeek-AI (2025); Yang et al. (2025); Team (2025b); Team et al. (2025). This breakthrough renders LLMs viable for tackling complex real-world problem. Drawing inspiration from the fundamental topological dependencies of communication network infrastructure, we introduce an innovative methodology for unified data modeling and generation. As shown in Figure 1, we integrates raw alerts and topological resources via a knowledge graph, and then build a root cause inference graph.

Unlike common reasoning tasks, alarm-based RCA is fundamentally a multi-label classification problem over complex, graph-structured data. A promising approach to automate this process involves integrating the structured knowledge representation of Knowledge Graphs (KGs) with the

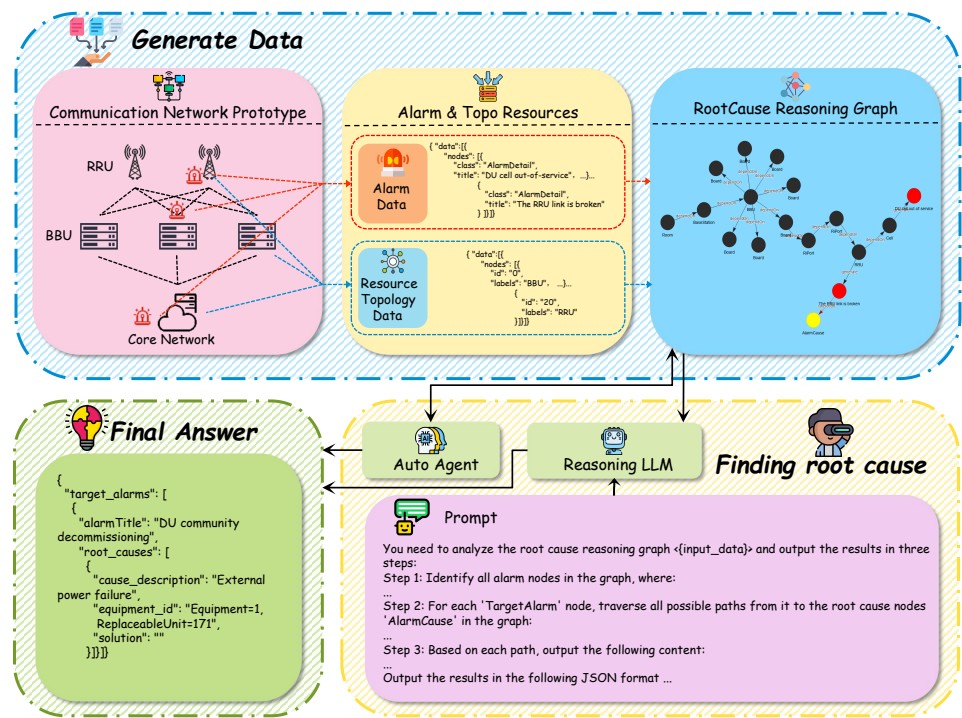

Figure 1: The framework for LLM-based root cause analysis in telecommunication base stations. The process begins by modeling real-world base station equipment (e.g., RRU, BBU) and their connections. The RootCause Reasoning Graph, along with expert-defined prompts, is fed into the Reasoning LLM or Auto Agent to perform root cause analysis.

reasoning capabilities of LLMs. As illustrated in Figure 1, the root cause reasoning graph is entered into the Reasoning LLM to directly leverage its reasoning power for root cause analysis. Despite its conceptual promise, the practical development of AI for Root Cause Analysis (RCA) in telecommunication network has been hindered by a salient gap: the lack of domain-specific benchmarks. This gap prevents a true assessment of LLMs' capabilities against unique challenges like 5G-induced alarm storms and complex network dependencies. To quantify this challenge, we build **TN-RCA**, the first root cause analysis benchmark in telecommunication network. Then we evaluate the prominent LLMs on the TN-RCA and get a sobering result. we find that LLMs' inherent reasoning ability hits a performance ceiling (58.99% F1-score) on this challenging benchmark, proving the need for specialized approaches beyond simple model scaling.

To bridge this gap, we present **Auto-RCA**, a novel agentic framework that demonstrates a path to mastery on this benchmark. Rather than enhancing the LLM's inherent abilities, Auto-RCA leverages the LLM as a brain in the agentic framework. This framework iteratively refines a code-based solution by systematically analyzing patterns across all failures.

Our work makes the following key contributions:

- **A Novel Telecommunication Network RCA Benchmark (TN-RCA):** We introduce the first real-world benchmark for RCA of telecommunication networks, featuring 530 scenarios grounded in expert-verified knowledge graphs.

- **An Agentic Framework with a Contrastive Feedback Loop (Auto-RCA):** We design Auto-RCA, a modular agentic system that iteratively improves a code-based solution. Its core innovation is a novel "evaluate-analyze-repair" loop that generates powerful, structured contrastive feedback by systematically analyzing aggregate failure patterns. This guides the LLM to perform targeted, hypothesis-driven code generation and repair.

- **State-of-the-Art Problem-Solving Capability:** Through extensive experiments, we show that the Auto-RCA dramatically improves the ability to solve the tasks in our benchmark,

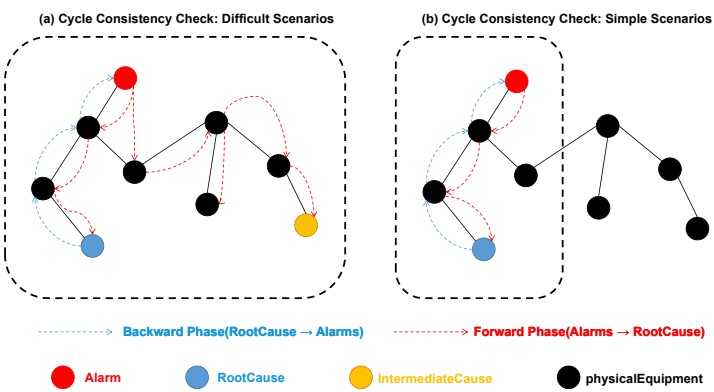

Figure 2: Illustration of the cycle consistency check for automated difficulty grading. (a) **Difficult Scenario**, where the forward reasoning phase (red arrows, Alarms → RootCause) results in a one-to-many mapping ($|m| > 1$). (b) **Simple Scenario**, where the forward mapping is one-to-one ($|m| = 1$). The causal path from the Alarm to the RootCause is unambiguous, and the backward phase (blue arrows, RootCause → Alarms) confirms this unique and satisfies cycle consistency.

elevating the final F1-score on TN-RCA to **91.79%**. This establishes a new state-of-the-art performance on this specific problem and charts a viable path toward automating complex, domain-specific reasoning. Meanwhile, Auto-RCA achieves **50%** accuracy on the Open-RCA Xu et al. (2025) benchmark and demonstrates its generalization.

## 2 METHODOLOGY

Our methodology addresses the challenge of telecommunication root cause analysis through a two-part contribution. First, we introduce **TN-RCA**, a novel and challenging benchmark grounded in real-world data. Second, we present **Auto-RCA**, an agentic framework designed to iteratively learn and master the complexities of the benchmark. This section details the construction principles of TN-RCA benchmark and the architecture of the Auto-RCA system.

### 2.1 BENCHMARK CONSTRUCTION: THE TN-RCA BENCHMARK

The primary contribution of our work is TN-RCA, a benchmark comprising 530 distinct tele-network failure scenarios. Its construction was governed by four core principles: **Veracity, Comprehensiveness, Verifiability, and Complexity Discriminability**. To achieve these principles, we adopted a result-oriented construction process, which we detail below.

**Veracity**: To ensure the benchmark reflects real-world challenges, all data is sourced directly from operational telecommunication base stations. This includes the network topology, which describes the physical connections between equipment (e.g., Baseband Units (BBUs), Remote Radio Units (RRUs)), and the alarm datas, which are real signals generated during actual fault events.

**Comprehensiveness**: We employ a result-oriented methodology. We first identify a comprehensive set of known root causes based on industry standards and expert knowledge. Then we start from these root causes, tracing their potential alarm manifestations through the network topology. This ensures a diverse coverage of root cause types, capturing both common and long-tail fault scenarios.

**Verifiability** : The result-oriented approach provides an inherent ground truth for each scenario. Since every case is constructed by tracing from a known, expert-verified root cause to its resulting alarms, the correct answer (the root cause) is embedded in the data by design. This provides a reliable and unambiguous basis for evaluating model performance.

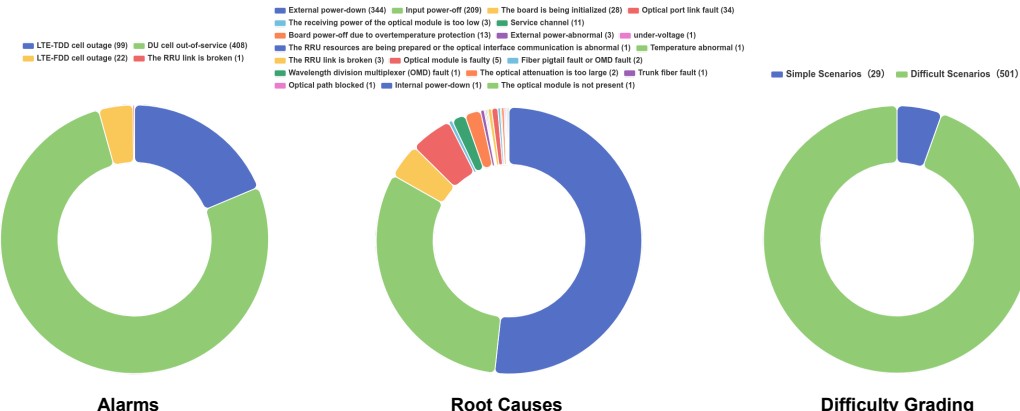

Figure 3: Distribution and Complexity Stratification of Root Causes in TN-RCA. The chart displays the frequency of major root cause categories, illustrating the benchmark's long-tail distribution. Each bar is segmented to show the absolute number of 'Simple' versus 'Difficult' scenarios, highlighting the benchmark's high level of challenge.

**Complexity Discriminability**: A key innovation of our methodology is the ability to objectively and automatically stratify the difficulty of scenarios. This is achieved through a cycle consistency check between a backward and a forward reasoning process, grounded in the graph structure.

- **Backward Phase (Root Cause → Alarms):** This phase establishes the ground truth. For a given standard root cause (e.g., "External Power Failure"), we utilize the resource topology graph and expert-defined rules to determine the complete chain of alarms that this cause would trigger. This creates a definitive mapping:

$$M_{backward} : RootCause \rightarrow \{Alarm_1, Alarm_2, \ldots, Alarm_n\} \quad (1)$$

- **Forward Phase (Alarms → Root Cause):** This phase simulates the LLM's task. Starting from the set of alarms identified in the backward phase, we then apply graph-based reasoning to generate a potential inference graph that links the alarms back to one or more possible root causes. This establishes a forward mapping:

$$M_{forward} : Alarms \rightarrow \{InferredRootCause_1, \ldots, InferredRootCause_m\} \quad (2)$$

The uniqueness of causal paths in the graph provides a theoretical basis for this process.

- **Cycle Consistency Check & Difficulty Grading:** We assess consistency by comparing the output of the forward phase with the input of the backward phase.
  - **Simple Scenarios:** A scenario is classified as "simple" if there is a one-to-one mapping ($|m| = 1$) and the inferred root cause perfectly matches the original root cause.
  - **Difficult Scenarios:** A scenario is classified as "difficult" if there is a one-to-many mapping ($|m| > 1$), where the alarms could plausibly point to multiple root causes, including the ground truth and several distractors. The difficulty level is proportional to the number of plausible alternative root causes.

This automated process removes subjective manual labeling of difficulty and provides a principled mechanism for analyzing model performance across a spectrum of complexity.

## 2.2 DATA REPRESENTATION AND STATISTICS

The TN-RCA benchmark encapsulates each scenario in two JSON files: *input.json* (RootCause Reasoning Graph) and *label.json* (ground-truth answer).

**Knowledge Graph Structure:** The *input.json* file models a fault scenario as a directed graph $G = (V, E)$, where $V$ is a set of nodes and $E$ is a set of edges representing their relationships. The nodes $V$ are primarily categorized into three types, essential for the RCA task:

- **Resource Topology Nodes:** These nodes represent the physical and logical components of a telecommunication base station, such as the Base Station (*BaseStation*), Baseband Unit (*BBU*), specific functional boards (*Board*, e.g., VBPe5a), Remote Radio Unit (*RRU*), and ports (*RiPort*). Each node contains properties like its logical distinguished name (*ldn*) and serial numbers, forming the structural backbone of the network.

- **Alarm Nodes (*AlarmDetail*):** These nodes represent the symptom data. They are linked to specific resource nodes, indicating which piece of equipment reported an alarm. Each alarm node contains rich metadata, including the *title* (e.g., "LTE Cell Out of Service"), *code*, *severity*, and the exact *reportAlarmTime*.

- **Root Cause Nodes (*AlarmCause*):** These nodes represent the potential root causes of the alarms. The LLM's goal is to identify the true root cause for a given *TargetAlarm*.

The edges $E$ in the graph define the relationships between these nodes. They include *dependOn* edges to describe the physical or logical hierarchy of the equipment, *generate* edges linking alarms to the equipment that produced them, and *causedBy* edges that represent causal hypotheses between different alarms or between a root cause and an alarm.

**Benchmark Statistics:** A statistical analysis of TN-RCA reveals two core characteristics: a realistic root cause distribution and a challenging design dominated by "difficult" scenarios.

First, the distribution of root causes in the benchmark exhibits a typical "long-tail" pattern (as shown in Figure 3). A few common faults (e.g., "External power-down") account for the majority of cases, while a large number of diverse. This distribution aligns closely with real-world operational data.

Second, we applied an automated "cycle consistency check" methodology to stratify all 530 scenarios by difficulty. TN-RCA includes 29 "Simple" scenarios and 501 "Difficult" scenarios. This means 94.5% of the cases are classified as "Difficult." This design establishes TN-RCA as a highly challenging benchmark capable of effectively differentiating the upper limits of various LLMs' reasoning abilities, especially when handling complex and ambiguous problems that remain unsolved.

## 2.3 THE AUTO-RCA AGENTIC FRAMEWORK

After establishing the TN-RCA benchmark to quantify the challenges of telecom RCA, we now introduce **Auto-RCA**, the agentic framework developed to master this task. Auto-RCA is not designed to enhance the LLMs' intrinsic reasoning but to orchestrate its capabilities within a iterative process that refines a programmatic, code-based solution by learning from its failure cases.

**Core Philosophy:** The central tenet of Auto-RCA is to treat failure not as an isolated incident but as a systemic learning opportunity. Instead of fixing bugs one by one, the framework analyzes patterns in all failure cases to identify the main logical deficiencies of the current solution code.

**System Architecture:** As illustrated in Figure 4, Auto-RCA employs a modular architecture composed of five synergistic modules that emulate an expert software development team. This structure separates concerns and ensures a robust, repeatable workflow.

- **The Orchestrator (Project Manager):** Manages the end-to-end "evaluate-analyze-generate-validate" lifecycle. It controls the workflow and makes the final decision to accept or reject a proposed code modification based on the failure cases.

- **The Evaluator (Test Engineer):** Quantifies the performance of a given code solution. It runs the code against another training set homologous to TN-RCA, calculates a precise F1-Score, and compiles all incorrect outputs into a set of Bad Cases for analysis.

- **The Bad Case Analyzer (Senior Analyst):** This is the intelligent core of the framework. It receives all Bad Cases from the Evaluator and performs a systematic analysis to identify and categorize the most prevalent failure mode (e.g., "missing root cause" vs. "extra root cause"). This analysis is the foundation for the system's contrastive feedback.

- **The LLM Agent (Coder & Thinker):** This module acts as the interface to the Large Language Model. It translates the structured output from the Bad Case Analyzer into a high-quality, targeted prompt. This prompt steers the code generation process of **Auto-RCA** toward a specific, logical fix.

Figure 4: The architecture of the Auto-RCA agentic framework. The system operates in a continuous loop where the Orchestrator manages the workflow between modules dedicated to evaluating code performance, analyzing failures, generating targeted prompts for an LLM, and sanitizing the resulting code for the next iteration.

- **The Sanitizer (Code Reviewer):** Ensures the reliability of the LLM's output. It cleans and formats the generated text to produce a pure, syntactically correct Python file, removing conversational artifacts or inconsistencies before the code is passed to the Evaluator.

**Iterative Workflow:** The Auto-RCA operates in a closed loop designed for progressive refinement:

1. **Baseline Testing (Round 0):** An initial code version is run on the training set, establishing an initial F1-Score and generating the first set of Bad Cases.

2. **Failure Analysis:** The Orchestrator passes the Failure Cases to the Bad Case Analyzer, which identifies the most critical systemic flaw.

3. **Guided Code Generation:** The LLM Agent uses this analysis to construct a targeted prompt and queries the LLM for a new, improved version of the solution code.

4. **Evaluation & Decision:** The candidate code is sanitized and re-evaluated on the benchmark. The Orchestrator compares the new F1-score to the previous best. If the score improves, the new code is accepted and becomes the basis for the next iteration.

5. **Iteration:** The process repeats, using the new set of (fewer) Bad Cases from the successful run as input for the next round of analysis, progressively refining the solution.

## 2.4 EVALUATION FRAMEWORK AND METRICS

To systematically measure LLMs' performance on TN-RCA, we developed a comprehensive evaluation framework and a set of precise metrics.

**Evaluation Protocol:** Our framework is designed for automated execution. The framework presents each of the 530 scenarios as an input knowledge graph. The LLM is asked by a structured set of instructions to analyze the graph and return its findings in a standard JSON format.

**Primary Metric (F1-Score):** Given the multi-label nature of the task (i.e., multiple alarms may be present, and a single alarm could have multiple root causes), we adopt the F1-Score as primary metric. It provides a balanced measure of a LLM's ability to correctly identify all true root causes. The metric is calculated as follows:

- **Extraction:** The evaluation protocol parses the generated JSON to extract the set of predicted root causes ($P$). Then it extracts the ground-truth set of root causes ($T$) from the corresponding label file.

- **Matching:** A predicted root cause $p \in P$ is considered a True Positive (TP) if it exactly matches a ground-truth root cause $t \in T$.

- **Calculation:** We compute Precision and Recall:

$$\text{Precision} = \frac{|P \cap T|}{|P|} \qquad\qquad \text{Recall} = \frac{|P \cap T|}{|T|}$$

- The F1-Score is the harmonic mean of Precision and Recall:

$$F_1 = 2 \times \frac{\text{Precision} \times \text{Recall}}{\text{Precision} + \text{Recall}}$$

The final reported score is the macro-average of the F1-Scores across TN-RCA benchmark. Our framework also reports overall Precision and Recall to provide deeper insights.

## 3 EXPERIMENTS

Table 1: Performance Breakdown by Scenario Difficulty on TN-RCA. The best-performing value in each column is highlighted in **bold**.

| Model | Precision@1 | | | Recall@1 | | | F1-Score@1 | | |
|---|---|---|---|---|---|---|---|---|---|
| | **Simple** | **Difficult** | **Mixed** | **Simple** | **Difficult** | **Mixed** | **Simple** | **Difficult** | **Mixed** |
| DeepSeek-R1-671B | 0.8448 | 0.4240 | 0.4471 | 0.8448 | 0.9600 | 0.9537 | 0.8448 | 0.5749 | 0.5897 |
| Gemini-2.5-Pro | **0.8621** | 0.4311 | 0.4356 | **0.8621** | 0.9658 | 0.9134 | **0.8621** | 0.5824 | 0.5899 |
| Claude-3.5-Sonnet | **0.8621** | 0.4021 | 0.4234 | **0.8621** | 0.9653 | 0.9537 | **0.8621** | 0.5677 | 0.5934 |
| Claude-3.7-Sonnet | **0.8621** | 0.4370 | 0.4441 | **0.8621** | 0.9800 | 0.9626 | **0.8621** | 0.5912 | 0.6078 |
| Claude-Sonnet-4 | **0.8621** | 0.4323 | 0.4561 | **0.8621** | 0.9783 | 0.9586 | **0.8621** | 0.5995 | 0.6181 |
| Qwen3-4B | 0.6752 | 0.3680 | 0.3782 | 0.7040 | 0.8302 | 0.8038 | 0.6741 | 0.4955 | 0.5144 |
| Qwen3-32B | 0.8302 | 0.4277 | 0.4498 | 0.8563 | 0.9756 | 0.9691 | 0.8311 | 0.5789 | 0.6142 |
| Qwen-QWQ | **0.8621** | **0.4381** | **0.4613** | **0.8621** | 0.9744 | 0.9682 | **0.8621** | 0.5900 | 0.6248 |
| Qwen3-235B-A22B | **0.8621** | 0.4374 | 0.4607 | **0.8621** | **0.9810** | **0.9736** | **0.8621** | **0.6049** | **0.6254** |

This section presents a series of experiments designed to answer two core research questions:

1. What is the upper performance bound of state-of-the-art LLMs on the TN-RCA benchmark when applied directly, relying solely on their inherent reasoning capacities?

2. Can Auto-RCA effectively improve the performance on the TN-RCA benchmark?

3. What is the generalization of Auto-RCA when evaluated on other benchmarks?

### 3.1 EXPERIMENTAL SETUP

- **Evaluation Metrics:** The primary evaluation metric is the macro-averaged F1-score, calculated across all 530 scenarios. To provide a more comprehensive performance view, we also report Precision@k (P@k) and Recall@k (R@k), where k=1.

- **Baseline Models:** To establish a robust baseline, we evaluated a diverse suite of prominent open-source and closed-source LLMs. As shown in Table 1, this includes models from the DeepSeek and Qwen series, as well as Google's Gemini, and Anthropic's Claude families.

- **Agent Configuration:** For the Auto-RCA, we selected several leading LLMs that support the long context required by the framework (at least 64K tokens). Each experimental run consists of five iterative optimization rounds, starting from the same initial code.

## 3.2 RQ1: BASELINE PERFORMANCE OF LLMS ON TN-RCA

To answer RQ1, we established a baseline by measuring the performance of various LLMs on TN-RCA benchmark. This experiment assesses their intrinsic reasoning capabilities when directly tasked with solving 530 real-world fault scenarios without any iterative refinement.

As shown in Table 1, the results clearly expose the limitations of current direct-application methods. On the "Simple" scenarios, many state-of-the-art LLMs performed capably, with LLMs like Gemini-2.5-Pro achieving a strong F1-score of **0.8621**. However, performance plummeted when faced with the "Difficult" scenarios. Consequently, the overall performance on the mixed dataset is modest. Qwen3-235B-A22B Team (2025a) achieved a top F1-score of only **0.6254**. This performance ceiling underscores the inherent difficulty of the TN-RCA benchmark and confirms that simply scaling general-purpose LLMs is insufficient to master the complex domain.

Table 2: Final Performance of Best Solution Code on the Mixed Dataset. Best value is in **bold**.

| LLM in Auto-RCA | P@1 | R@1 | F1@1 |
|---|---|---|---|
| DeepSeek-R1-671B | 0.8115 | 0.9288 | 0.8450 |
| Gemini-2.5-Pro | **0.8951** | 0.9767 | **0.9179** |
| Claude-3.5-Sonnet | 0.8097 | 0.8633 | 0.8262 |
| Claude-3.7-Sonnet | 0.8544 | 0.7656 | 0.7922 |
| Claude-Sonnet-4 | 0.4685 | **0.9823** | 0.6140 |
| Qwen3-235B-A22B | 0.6682 | 0.6626 | 0.6632 |

Table 3: Performance of different methods on OpenRCA Xu et al. (2025). Best value is in **bold**.

| LLM | Method | Category | | |
|---|---|---|---|---|
| | | Easy | Mid | Hard |
| | Oracle | 8.72 | 3.50 | 0.00 |
| Claude-3.5-Sonnet | RCA-agent | 16.78 | 9.09 | 0.00 |
| | **Auto-RCA(ours)** | **40.14** | **15.42** | **10.24** |
| | Oracle | 9.40 | 4.90 | 0.00 |
| Gemini-2.5-Pro | RCA-agent | 13.42 | 6.99 | 0.00 |
| | **Auto-RCA(ours)** | **50.06** | **20.34** | **17.68** |

## 3.3 RQ2: EFFECTIVENESS OF THE AUTO-RCA FRAMEWORK

To address the shortcomings identified in RQ1, we evaluated the effectiveness of our Auto-RCA framework. This framework recasts the problem from a single-shot inquiry to an iterative, code-refinement process. It employs an LLM as a reasoning engine to propose solutions in the form of code. These solutions are then automatically evaluated against the benchmark, and structured feedback is generated from failures to guide the LLM in the next iteration.

This agentic approach proved highly effective. As detailed in Table 2, the solution generated using Gemini-2.5-Pro within the framework reached a final F1-score of **0.9179**. More detailed experimental results are presented in the Supplementary Material.

A key factor in Gemini-2.5-Pro's superior performance was its large 1M token context window. This enabled the agent to maintain a comprehensive history of the code, past errors, and feedback, preventing critical information loss that hindered models with smaller context capacities.

The final result of **0.9179** is not just a score increase; it establishes a new state-of-the-art for this task. It validates our hypothesis that an autonomous, self-optimizing agentic architecture can effectively bridge the gap between general-purpose LLM capabilities and the nuanced demands of complex, domain-specific problem-solving like telecom RCA.

### 3.4 RQ3: Generalization of the Auto-RCA Framework

Auto-RCA, driven by higher-order LLMs, demonstrates markedly better and consistent performance across all difficulty levels: for example, on Claude-3.5-Sonnet Auto-RCA achieves **40.14/15.42/14.22** (Easy/Mid/Hard) versus RCA-agent's 16.78/9.09/0.00, and on Gemini-2.5-Pro it obtains **50.06/20.34/17.68** compared with 13.42/6.99/0.00; notably, Auto-RCA is the only method to produce substantial nonzero scores in Hard cases, indicating strong generalization and robustness.

## 4 RELATED WORK

### 4.1 KNOWLEDGE GRAPH ENHANCED LLMS

Integrating Knowledge Graphs (KGs) with Large Language Models (LLMs) is a key strategy to improve factual grounding and mitigate hallucinations Ibrahim et al. (2024); Dehal et al. (2025). Current research focuses on frameworks that align LLM reasoning with KG paths Shen et al. (2025b), use LLMs as policy models to navigate KGs Shen et al. (2025a), or leverage KGs to rerank generated candidates Ji et al. (2024). These approaches highlight a trend towards structured neuro-symbolic reasoning, which is essential for complex, fact-based tasks like RCA Amayuelas et al. (2025).

### 4.2 ROOT CAUSE ANALYSIS (RCA)

AI-driven RCA increasingly replaces manual methods by leveraging multi-modal data from logs, metrics, and traces Pham et al. (2025); Roy et al. (2024); Xu et al. (2025). While general benchmarks like Defects4J Just et al. (2014) and RCAEval Pham et al. (2025) have been vital, they do not address the unique challenges of telecommunication networks. RCA in telecoms is distinct due to massive "alarm floods" Zhang et al. (2021), complex topological dependencies, and stringent real-time constraints Li et al. (2023); Jakobson & Weissman (1995). Despite advances in applying AI/ML to this domain Zhang (2023); Ma et al. (2024); Wang et al. (2024); Liang et al. (2020); Bouloutas et al. (1994), the field has lacked a large-scale, public benchmark like our TN-RCA.

### 4.3 LLM-BASED AGENTS FOR AUTOMATED PROGRAM GENERATION AND REPAIR

The paradigm of using LLMs as autonomous agents for Automated Program Repair (APR) has shown significant promise, with benchmarks like SWE-bench becoming a standard for evaluation Campos et al. (2025); Ma et al. (2025). A prominent architectural trend is the use of modular agentic systems, where specialized sub-agents collaborate to solve complex tasks, a philosophy adopted by frameworks like MASAI Arora et al. (2024).

A critical aspect of these agents is the mechanism for self-correction. Early approaches like Self-Refine Pan et al. (2024), which rely on an LLM's intrinsic ability to fix its own output, have proven ineffective for complex reasoning tasks due to "cognitive fixedness" Kamoi et al. (2024). The consensus is that effective improvement requires reliable external feedback. Building on this, methods like ContrastRepair provide contrastive feedback using pairs of failing and passing tests to guide the LLMs Kong et al. (2024). Auto-RCA advances beyond instance-level fixes by using holistic failure analysis to generate strategic feedback, enabling LLMs to correct systemic flaws through a structured process that surpasses direct application.

## 5 CONCLUSION

This paper makes two core contributions to AI-driven network management. First, we introduced TN-RCA, a public, real-world benchmark that provides a standardized platform to measure and drive progress in telecom RCA. Second, we proposed Auto-RCA, a novel agentic framework that demonstrates a clear path to mastering this benchmark. The central insight of our work is that for complex, domain-specific reasoning tasks, the most effective paradigm may not be the direct application of LLMs. Instead, superior performance can be achieved by leveraging them as components within autonomous agentic frameworks that iteratively refine verifiable solutions. Future work will focus on extending the benchmark with greater complexity and generalizing the Auto-RCA framework to other challenging, graph-based problem domains.

ETHICS STATEMENT

We focus on ethical AI research and strive to achieve a balance between technological advancements and our ethical responsibilities. TN-RCA is released as a public, expert-validated benchmark built from rigorously anonymized telecommunication alarm data and provided for non-commercial research under an explicit license. Auto-RCA is intended solely for academic evaluation and methodology development, any real-world use should be preceded by thorough safety, reliability, and domain-specific risk assessments with human-in-the-loop oversight.

REPRODUCIBILITY STATEMENT

All data and experiments reported in this work are real and traceable; a detailed description of TN-RCA and the complete implementation of Auto-RCA are provided in the supplementary materials. The supplement includes data provenance and licensing information (including anonymization procedures and usage restrictions), raw telemetry together with knowledge-graph construction and preprocessing scripts, data splits and example entries, the full codebase (training, inference, and evaluation scripts), hyperparameter configurations, random seeds, and dependency and environment specifications (e.g., Conda/requirements or Docker configuration).

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
