# OpenReview forum: "TN-AutoRCA: Benchmark Construction and Agentic Framework for Self-Improving Alarm-Based Root Cause Analysis in Telecommunication Networks"
_ICLR.cc/2026/Conference — Submitted to ICLR 2026_

### Official Review · Reviewer_YSw8 · 2025-10-28

**Soundness:** 2
**Presentation:** 1
**Contribution:** 2
**Rating:** 2
**Confidence:** 5

**Summary:**

The paper explores using AI agents for root-cause analysis of failures in telecommunication networks. It presents a dataset called TN-RCA which comprises 540 failure scenarios that are synthesized from real-world data. The paper also presents an agentic solution called Auto-RCA which generates code to analyze root causes.

**Strengths:**

A useful failure dataset of telecommunication networks.

**Weaknesses:**

It is unclear how representative the dataset is regarding telecommunication networks. There are little details on the paper with regards to the comprehensiveness or representativeness of the dataset. For example, does the dataset include all the failure patterns you see in production telecommunication networks? Can you justify it?

The paper says that the dataset is sourced directly from the base stations. However, modern telecommunication networks is a complex networked system including edge presences and 5G telco clouds. If the dataset is collected from the base station, how can it include various faults and failure scenarios in the telecom backbone?

I don’t understand why you will always need to generate code for analysis. Most RCA agents can infer root causes from observability data. Have you tried those baseline approaches? If not, why?

I also do not understand what is the definition of root causes. From Figure 3, it seems that root causes in the paper mean hardware faults? What about software bugs?

Given that you need to generate code, it seems to me that Claude models generally perform better than the other models, because their codegen ability is higher, not because they have stronger reasoning ability for RCA.

There is a large body of work on using AI for RCA and site reliability engineering in general. The paper does not position itself very well. In fact, I have a hard time understanding what is the fundamental difference between telecom networks versus cloud systems and networks, from an AI perspective.

The presentation is poor. There are quite a few typos and grammar errors. The term “fault” and “failures” are used interchangeably, which is incorrect (see Laprie’s fault-error-failure model).

**Questions:**

Please see my questions in the Weakness column (I asked a lot).

---

### Official Review · Reviewer_GnVn · 2025-10-31

**Soundness:** 3
**Presentation:** 2
**Contribution:** 2
**Rating:** 2
**Confidence:** 3

**Summary:**

This paper first proposed a real-world benchmark TN-RCA constructed from expert-validated knowledge graphs. Then it proposed a multi-agent framework Auto-RCA that employs techniques like self-correction and iterative loop with code generation and excuetion. Experiments show the effectiveness of the proposed Auto-RCA.

**Strengths:**

S1: The proposed benchmark is the first RCA benchmark in telecommunication networks.

**Weaknesses:**

W1: The paper lacks focus. It proposed a new benchmark TN-RCA, which should be challenging. And then it proposed a new method Auto-RCA, which achieves a 91% F1-score. It seems that the new benchmark is not challenging enough.
W2: For the proposed benchmark TN-RCA, the authors only evaluated LLM-based methods, but ignored the traditional RCA methods. RCA for graph-structured data has been extensively studied in the microservice domain [1][2][3]. It is not clear how they perform on these benchmarks. I am not sure whether they can be directly applied to the Telecommunication Network domain. But I am confident there are other works in Telecommunication Network literatures working on RCA.
W3: The contribution and novelty of Auto-RCA is not clear. The authors applied a multi-agent framework to implement Auto-RCA. But it is not clear what the challenge is how they solve it. The conclusion that agentic framework works better than pure LLM brings no new knowledge to the community.

[1] MicroHECL: High-Efficient Root Cause Localization in Large-Scale Microservice System. ICSE-SEIP 2021
[2] Sage: Practical and Scalable ML-Driven Performance Debugging in Microservices. ASPLOS 2021
[3] ShapleyIQ: Influence Quantification by Shapley Values for Performance Debugging of Microservices. ASPLOS 2023.

**Questions:**

See weaknesses

---

### Official Review · Reviewer_k3kc · 2025-10-31

**Soundness:** 3
**Presentation:** 2
**Contribution:** 2
**Rating:** 2
**Confidence:** 4

**Summary:**

This paper presents TN-RCA, a benchmark for alarm-based root cause analysis (RCA) in telecommunication networks, and Auto-RCA, an agentic framework that iteratively improves code-based RCA performance through contrastive feedback. The benchmark stems from real-world data and the results show notable improvements over direct LLM reasoning. However, the paper lacks supporting details such as the training data statistics, case analyses, and baseline comparisons. It remains unclear how the proposed framework is specifically related to the telecommunication setting, as the method appears generally applicable to any code-based reasoning task.

**Strengths:**

1. The benchmark and experiments are grounded in real-world telecommunication scenarios and data.
2. The paper demonstrates that a code-based solution can be effective for RCA tasks, achieving strong results on OpenRCA.

**Weaknesses:**

1. Novelty and relevance: The Auto-RCA framework appears applicable to general code-based problem-solving tasks; it is unclear which design elements are specifically tailored to the telecommunication RCA context.
2. Missing statistics: The paper lacks details about the training set’s size and difficulty distribution, which are crucial given that Auto-RCA’s core mechanism relies on contrastive feedback.
3. Missing case analysis: The paper is missing failure cases of LLM-only methods and pre-/post-optimization code, which would help illustrate the improvement process.
4. Baseline comparison: The experiments do not include other code-based baselines, making it difficult to assess Auto-RCA’s relative advantage.

**Questions:**

1. What is the size of the training set, and what is its difficulty distribution?
2. Authors stated in Related Work 4.2 that Alarm floods, complex topological dependencies, stringent real-time constraints are the unique challenges of telecommunication networks. How does the TN-RCA benchmark address these challenges?
3. What is the major difference between TN-RCA and OpenRCA in terms of data design, scope, and evaluation methodology?
4. Can the author provide some case analysis on failures and optimized codes?

---

### Official Review · Reviewer_FxuV · 2025-11-01

**Soundness:** 3
**Presentation:** 2
**Contribution:** 2
**Rating:** 4
**Confidence:** 5

**Summary:**

This paper introduces TN-RCA, the real-world benchmark for root cause analysis (RCA) in telecommunication networks, comprising 530 expert-verified scenarios. It also proposes Auto-RCA, an agentic framework that leverages LLMs within an iterative “evaluate-analyze-repair” loop to improve code-based RCA solutions. The authors demonstrate that while state-of-the-art LLMs plateau at 58.99% F1-score on TN-RCA, Auto-RCA elevates performance to 91.79%, establishing a new state-of-the-art. The framework also shows strong generalization on the OpenRCA benchmark.

**Strengths:**

1. New Benchmark: TN-RCA is the benchmark for RCA in telecom networks, addressing a gap in the field. It is built on real-world data and features a principled difficulty stratification method via cycle consistency checks.
2. Agentic Framework Design: Auto-RCA introduces a modular, feedback-driven agent architecture that systematically improves code-based solutions.

**Weaknesses:**

1.  Limited Motivational Depth: While RCA is an important industrial problem, the paper does not sufficiently justify why it is a meaningful or challenging AI research task. The connection to broader machine learning challenges—beyond telecom-specific applications—is underdeveloped, making the contribution feel more applied than foundational.
2.  Superficial Experimental Analysis: The experiments, though comprehensive in model comparisons, lack in-depth analysis. For instance, there is little discussion of why certain LLMs fail on difficult cases, what types of logical errors are common, or how the agent’s feedback loop specifically addresses these failures. This limits insight into the mechanics of both the benchmark and the framework.

**Questions:**

1.  Regarding the RCA task, why is it not positioned as a novel task paradigm that sits at the intersection of LLMs and graph learning? The current version of the paper reads more like the discovery of a complex new dataset in an industrial setting, where you observed that LLMs struggle, and subsequently proposed a benchmark. However, I am more concerned with the broader impact of your benchmark on the general LLM research community. In its present form, the influence of your benchmark seems limited to researchers specifically interested in this narrow task (or in 5G), which, I believe, constrains the significance of the contribution.
2.  Does the construction of the TN-RCA benchmark from TELECOMMUNICATION NETWORKS data raise any potential user privacy leakage issues? Could the authors please clarify the data anonymization procedures and safeguards implemented?
3.  Concerning the experimental section, the number of experiments conducted seems insufficient. For RQ1, RQ2, and RQ3, the experiments lack comprehensiveness, and the depth of the experimental analysis is inadequate. Furthermore, I did not see any concrete case studies that illustrate how the framework succeeds or fails in specific instances, which would provide crucial qualitative insights.

---

### Meta-Review · Area_Chair_zTa3 · 2026-01-04

**Summary:**

The reviewers pointed out significant issues about the clarity, relevance, and empirical evaluation of the manuscript. Since there was no rebuttal provided, these concerns remain unresolved. I therefore recommend rejection.

**Reviewer Concerns:**

No rebuttal was provided.

Outstanding concerns:
- Presentation is poor. Certain terms are not used or defined properly
- Limited explanation of the motivation/relevance and novelty
- Empirical studies are not sufficient (e.g., missing baselines, missing case analysis, lack in-depth analysis)

**Reviewer Scores:**

Not applicable because no rebuttal was provided.

---

### Decision · Program_Chairs · 2026-01-26

Reject